# A Clinical Trial of Entolimod a TLR-5 Adjuvant for Vaccines Using Diphtheria or Tetanus as Carrier Proteins

**DOI:** 10.3390/vaccines10101592

**Published:** 2022-09-22

**Authors:** Thomas R. Kosten, Coreen B. Domingo, Colin N. Haile, David A. Nielsen

**Affiliations:** 1Department of Psychiatry and Immunology, Baylor College of Medicine, Houston, TX 77030, USA; 2Department of Psychology, University of Houston, Houston, TX 77004, USA; 3Department of Psychiatry, Baylor College of Medicine, Houston, TX 77030, USA

**Keywords:** human vaccine, TRL5 agonist, diphtheria/tetanus, pharmacogenetics, anti-drug vaccine, vaccine adjuvant

## Abstract

Anti-drug vaccines previously failed clinical trials because they did not provide a sufficient titer or duration of antibodies (AB), but new adjuvants enhance both AB titers and efficacy duration. This clinical trial assessed AB titers after a single booster of commercial tetanus-diphtheria (Td) vaccine in 40 males randomized as 15 to Td alone and 25 to Td combined with the TLR5 adjuvant, Entolimod (Ent). Ent significantly increased ABs against diphtheria (DPT) (0.46 vs. 0.29 IU/mL increase; n = 40, *p* < 0.05), but against tetanus (TT) only if baseline TT AB was below 3 IU/mL (3.1 vs. 2.1 IU/mL; n = 20; *p* < 0.05). These 20 participants also showed a two-fold increase in anti-TT AB titer more often when given Ent than non-Ent (33% vs. 82%) (*p* < 0.03). Anti-Ent AB was low and appeared unlikely to reduce Ent efficacy after repeated Ent administration. Medical safety was excellent, and a *TLR5* missense polymorphism reduced anti-DPT AB production, but Ent increased anti-DPT AB titers to levels induced in subjects with genetically “normal” TRL5 functioning. Further clinical testing of TLR5 adjuvants like Ent seems warranted for anti-drug vaccines.

## 1. Introduction

Anti-drug vaccines are made by chemically attaching abused drugs like cocaine and fentanyl (haptens) to protein carriers such as tetanus toxoid (TT) and diphtheria (DPT) toxoid [1,2]. Vaccinating humans and animal models with this combination produces antibodies (AB) against these abused drug haptens. which are otherwise too small to induce AB responses [1,2]. Eliciting ABs is a complex process involving activation of both B cells and CD4 T cells, which the TT or DPT carriers alone can elicit and on which the abused drug haptens depend. When a fully vaccinated drug abuser then uses that particular drug (e.g., cocaine), the induced anti-drug AB then will attach to the abused drug. Because of the size of the AB attached to the drug, this drug-AB combination prevents the drug (e.g., cocaine) from entering the brain. Preventing brain entry of the cocaine then blocks the euphoria and reinforcement that otherwise would occur when ingesting the drug.

This procedure and mechanism of action for producing anti-drug ABs has worked well in animal models; however, none of the clinically tested anti-drug vaccines have provided a sufficient titer of blocking AB for efficacy in about one-third of the vaccinated subjects [1,2]. Duration of efficacy is also limited with many vaccines. For example, boosters are needed every seven to ten years for TT and DPT AB titers to remain protective, and haptenated vaccines against addictive drugs have required boosters about every 3 to 4 months to maintain adequate anti-drug AB titers for therapeutic drug blocking [3]. Less frequent boosters would be preferred [3].

This need to increase antibody titers above the threshold needed for efficacy can be overcome using alternative adjuvants in addition to the usual adjuvant of aluminum hydroxide (alum). Several other adjuvants have been tested with anti-addiction vaccines in animals including DNA products (CpG ODN 1826), Toll-like receptor (TLR) agonists like TLR4 agonists (MonoPhosphoLipid-A and E6020 from Eisai) or TLR5 agonists (flagellin) and others [4]. The TLR5 agonist flagellin and Ent have both enhanced the AB response to drugs of abuse previously in animals [5] We assess a new TLR5 adjuvant Entolimod (Ent) for its ability to increase the AB response to TT and DPT after a single booster of the standard Td vaccine in this clinical study of healthy human males. This research study was designed to inform a future clinical trial administering Ent combined with our newly developed anti-drug vaccines against cocaine, methamphetamine or opioids, which use TT or DPT as the carrier.

The potential new protein adjuvant Ent is derived from the bacterial endotoxin flagellin and is a TRL5 agonist [6]. Cleveland Biolabs Inc. (CBLI) produced Ent for use in oncology, and they have given it to normal humans and to cancer patients at doses up to 40 μg/day over 2-week periods without inducting cytokine storms or other serious toxicities [7,8]. This safety profile suggests that Ent can be combined with other immunotherapeutic agents at a dose of 30 μg/day. However, we have found that Ent markedly increased AB titers to both TT and to haptens such as amphetamine in rodents at a dose of 1 ug with no further increase in peak anti-hapten AB at doses up to 10 ug [9,10]. We therefore selected 1 ug, which is 30 to 40-fold lower than the previously safe doses in humans, and we tested the hypothesis that the AB titers against TT and DPT would be significantly greater in those participants given Ent plus tetanus-diphtheria (Td) vaccine compared to those given the Td vaccine alone in this clinical trial. We also assessed the anti-Ent AB titers in the subjects given Ent noting that CBLI considers anti-Ent AB levels above 3500 Units as having a potential for diversion of anti-drug AB production to anti-Ent AB production. Finally, we explored two potential pharmacogenetic impacts of a missense polymorphism in the TLR5 receptor on vaccine antibody response: 1. reducing the AB response to the Td vaccine and 2. testing whether Ent’s TLR5 agonism might increase the AB response to Td vaccine in participants with this polymorphism to match the response levels shown by participants without this missense mutation [11].

## 2. Methods

### 2.1. Study Design and Overview

This single-blind study included 40 healthy normal male participants randomly assigned into two groups in a 3:5 ratio with 15 receiving intramuscular Td vaccine alone and 25 getting Td vaccine + Ent (1 µg). The Td Adsorbed vaccine dose (0.5 mL) contained tetanus toxoid (5 Lf) and diphtheria toxoid (2 Lf) adsorbed on aluminum phosphate and suspended in isotonic sodium chloride solution (Mass Biologics, Mattapan, MA, USA). We measured IgG AB titers against TT, DPT and Ent at baseline before vaccination and six weeks after the vaccination. We expected that Td + Ent would double the anti-TT and anti-DPT AB titers over Td alone. Since Ent is derived from flagellin, we expected some cross-reactive AB against Ent at baseline as well as a further increase at week 6. Safety was monitored by the collection of vital signs, physical examinations, safety laboratory tests, and adverse events (AEs). Patients were excluded for current diagnosis of other drug or alcohol dependence (other than tobacco), acute or unstable medical illness, history of severe mental illness (such as psychosis, schizophrenia, or bipolar disorder), current suicidality, or inability to read and understand the consent form. This consent, as well as specific consent for genetic studies, was signed by all screened patients, and approved by Institutional Review board of Baylor College of Medicine.

### 2.2. Participants

From December 2017 to March 2018, we screened 96 participants and recruited 40 healthy normal male participants aged 28.6 years (range 18 to 39) with no significant differences between the Ent and non-Ent groups in age (28.1 vs. 28.8) or Caucasian ethnicity (53% vs. 32%, chi sq = 1.8). They had no chronic illnesses and were taking no medications. The included participants reported Td vaccinations 6 to 15 years previously, and anyone reporting Td vaccinations within the past 5 years was excluded to minimize high baseline titers of anti-TT and anti-DPH AB. Other exclusions were over 40 years old, being female, blood test abnormalities (elevated glucose or liver functions), substance abuser (amphetamine), serious illness or surgery within the past year and 10 were simply lost after the screening call.

### 2.3. Outcomes

We assessed IgG titers against TT, DPT and entolimod using enzyme-linked immunosorbent assays (ELISAs) from Alpha Diagnostic International (San Antonio, TX, USA) for TT and DPT and an ELISA from CBLI for entolimod. We used the iMark Microplate Absorbance Reader to assess the optical density of each cell on the plate. We finally determined sample concentrations from the resulting curve fit using Four Parameter Logistic (4PL) curve fit from symmetrical sigmoidal calibrators.

Safety was monitored by collection of vital signs, physical examinations, safety laboratory tests, and adverse events (AEs). We examined the triceps muscle vaccination site before and 8 h after intramuscular vaccination for AEs. We also assessed any signs of systemic reactions (including cytokine storm) to the vaccine during those 8 h. We assessed the primary symptoms of a cytokine storm including high fever, swelling, redness, extreme fatigue, and nausea. A medically trained staff member or physician reviewed AEs face to face with the subject weekly starting on the day of randomization (Study Day 0) through the End of Study Visit (Week 6) and through six months following the study vaccination for long-term follow-up.

We also explored the potential impact of a *TLR5* genetic missense polymorphism (*rs2072493*) on Ent’s efficacy, as a TLR5 agonist, for increasing the anti-TT or anti-DPT AB titers. As a polymorphism that attenuates the coupling of the TLR5 receptor to its intracellular effector system, we expected those individuals with this polymorphism to have a blunted or no effect from the Ent beyond the alum alone and have lower titer increases than those without this polymorphism and comparable titers to those who did not get Ent [9,11].

### 2.4. Genetic Testing

DNA was isolated from the blood as described previously [12]. DNA was purified from approximately 8 mL blood following the Gentra Puregene protocol (Qiagen, Valencia, CA, USA). PCR amplifications were performed using Platinum Taq DNA Polymerase High Fidelity (Invitrogen, Carlsbad, CA, USA) on a GeneAmp PCR system 9700 (Applied Biosystems). Genotypes were determined using a 5′-fluorogenic exonuclease assays (TaqMan, Applied Biosystems, Foster City, CA, USA). The *TLR5* variant *rs2072493* was genotyped using the TaqMan primer-probe sets (Applied Biosystems) assay ID C_22273027_10. PCR amplifications were performed using Platinum quantitative PCR SuperMix-UDG (Invitrogen, Carlsbad, CA, USA) on a ViiA 7 (Applied Biosystems). The *TLR5* gene is encoded on the reverse strand of the Human Genome Browser, therefore the variants referenced herein are those found on the reverse (i.e., sense) strand. Data analysis was evaluated with the ViiA 7 Software v1.1 (Applied Biosystems, Foster City, CA, USA). One Ent and one non-Ent subject declined to be genotyped.

### 2.5. Data Analyses

Our data analyses had two known complicating issues: 1. the wide range of AB responses to Td vaccine boosting and 2. the impact of baseline AB titers before boosting. The anti-DPT and anti-TT AB titers in normal adults at one or two months after boosting have very wide ranges. For anti-TT this range is 0.85–673.5 µg/mL (0.05–39.6 IU/mL, based on 1 IU/mL = 17 ug/mL) [13,14]. This wide range can be compressed by calculating a ratio of the boosted titers divided by the baseline anti-TT AB before the booster vaccination, which has yielded a more uniform 6-fold increase across participants [11]. Thus, we considered using the baseline anti-TT titers before the booster vaccination to calculate this ratio for comparison between the Ent and non-Ent groups on anti-TT and anti-DPT titers. However, a related complication is the actual values of baseline AB titers, which can be relatively high among participants who are not selected based on a current need for booster vaccination. Boosters for TT are typically given when AB titers fall between 0.1 to 0.5 IU/mL, and the Danilova study selected individuals needing such a booster. They found the mean increase in AB titer after a booster was 1.6 IU/mL and with a baseline average of 0.3 IU/mL, this yields an average ratio of a 6.3-fold increase (1.9/0.3), as reported by Danilova [13]. However, when baseline AB titers are substantially greater than 0.5 IU/mL, this average amount of increase in AB titer of 1.6 IU/mL after a booster will not provide ratios of 6-fold increases. With baseline anti-TT AB titers of 3 IU/mL, a doubling of AB titer to 6 IU/mL would require twice the average increase in actual AB titer compared to this previous standard from Danilova [13]. Furthermore, a 6-fold increase would require a titer increase to 18 IU/mL, which is beyond the standards for most commercial ELISA assays. Equivalent analyses of normal data for anti-DPT AB titers indicates that booster vaccination should be done when these AB titers range from 0.01 to 0.1 IU/mL, but our participants were not selected to have such low baseline AB titers. Thus, we relied on the comparison between the Ent and non-Ent groups using simple increases in anti-DPT AB titers from baseline rather than making any other adjustments for the baseline titers.

To reduce the problem of a ceiling effect on the change in TT or DPT AB titers from boosting, we used stratified analyses for comparing the two Ent groups. We also performed separate correlations between the baseline AB titers and the change in TT or DPT AB titers for the Ent and non-Ent groups. We then did a stratified comparison of the two Ent groups using the overall mean baseline AB titer for stratification, if these correlations were significantly different between the Ent groups. We also made this baseline stratification for anti-Ent AB.

For the genetic analyses, we combined the heterozygotic allele with the less frequent allele, because of the relatively small sample being examined and the expected allele frequencies. We therefore only tested the AA genotype versus the AG and GG participant groups for difference in response to Ent.

## 3. Results

The mean change in anti-DPT AB titers from a baseline of 0.64 IU/mL for Ent and 0.67 IU/mL for non-Ent was significantly greater for the Ent than non-Ent group (0.46 vs. 0.29 IU/mL; *p* < 0.05), and the correlation between baseline and the change in anti-DPT AB titers was greater for the Ent group (r = 0.96 vs. 0.72; Z = 2.9; *p* < 0.005), as shown in Figure 1A. We therefore stratified on the mean overall baseline of 0.7 IU/mL. This yielded 17 Ent and 8 non-Ent participants in the lower baseline AB groups and showed a greater increase for the Ent group (1.1 vs. 0.8 IU/mL; *p* < 0.01). The higher baseline AB group showed no significant difference in the AB rise between the two Ent groups (0.42 vs. 0.43 IU/mL; NS).

The mean change in anti-TT AB titers from the baseline means of 3.0 IU/mL for Ent and 2.9 IU/mL for non-Ent was not significantly different (1.81 vs. 1.82 IU/mL). The correlation between the baseline and change in anti-TT AB titers after vaccination was significantly larger for the Ent than non-Ent group (0.90 vs. 0.58; z = 2.3, *p* < 0.02), as shown in Figure 1B We therefore stratified on the mean baseline of 3 IU/mL anti-TT AB titer yielding 11 Ent and 9 non-Ent participants in the lower baseline AB groups and showed a greater increase for Ent (3.1 vs. 2.1 IU/mL; *p* < 0.05). This same stratification yielded significantly more Ent than non-Ent participants with a greater than two-fold increase in anti-TT AB titer (3/9 non-Ent vs. 9/11 Ent; chi sq = 4.3; *p* < 0.03). The higher baseline AB group showed no significant difference in the AB rise between the two Ent groups.

The mean anti-Ent AB levels in the 25 Ent treated men was 464 Units (range = 100 to 1400 Units) at baseline and 1402 Units at 6 weeks (range = 250 to 6800 Units), which was a significant three-fold rise (Paired *t*-test, *p* < 0.005). Eliminating two subjects with titers at week 6 of 3900 and 6800 units reduced the week 6 mean to 1058 Units and the AB rise to 2.2 fold; but statistical significance increased (Paired t-test, *p* < 0.0005). While three subjects had baseline anti-Ent titers above 1000 Units, the two subjects with high week 6 titers had baseline titers of 100 and 500 Units suggesting that baseline titers will not be a useful screening for eliminating these stronger producers of anti-Ent AB.

Our safety analyses of vital signs, physical examinations, safety laboratory tests, and adverse events (AEs) showed no serious AEs during the 6-week study or at a 6-month follow-up. Specifically, laboratory AEs at 6 weeks were more frequent in the non-Ent than the Ent group (9 vs. 2 AE) and included 5 cases of elevated glucose and two of an elevated liver enzyme in the non-Ent group. None of them were attributed to the Ent. We also assessed anti-Ent AB titers at baseline and 6 weeks. Participants had some detectable AB against Ent at baseline, although the cross-reactivity of Ent and flagellin is quite low and no subjects were previously exposed to Ent. The baseline anti-Ent AB titers were quite low, and these AB were not neutralizing against the Ent, although the anti-Ent AB titers rose 1.6-fold (range 1.1 to 5-fold) among the participants given Ent.

The genetic analyses for the *TLR5 rs2072493* polymorphism yielded 17 participants with an AA genotype and 7 participants with AG/GG genotypes for Ent and 10 AA and 4 AG/GG for non-Ent participants (no significant difference in AA rates). We found a significant effect of the Ent on increasing anti-DPH titers (F = 2.97; df = 3, 37; *p* < 0.05). The participants carrying the AG/GG polymorphism had 4-fold greater anti-DPT AB increases (0.08 vs. 0.47 IU/mL) with Ent (n = 7) than without Ent (n = 4), while those without this polymorphism showed no difference in anti-DPT AB increases with the addition of Ent (0.43 vs. 0.37 IU/mL). However, the anti-TT titers showed no difference between the two genotype groups with Ent (1.82 vs. 2.07 IU/mL) or without Ent (1.82 vs. 1.78 IU/mL). These DPT findings were consistent with our hypothesis of reduced AB production associated with a potentially defective TLR5 receptor and with compensation for this reduced AB production using a TLR5 agonist in these participants.

## 4. Discussion

This study showed a significant increase in boosted AB production against DPT and TT using the TRL5 agonist Ent as an adjuvant in 40 adult males. The DPT AB rise was 60% greater with Ent than with placebo Ent (non-Ent), but the TT AB rise showed no difference between Ent and non-Ent. Moreover, the baseline AB titers against TT and DPT had a critical impact on the subsequent rise in the AB titers after boosting, because our baseline TT AB titers were higher than the TT AB titers after boosting in one previous study of TT vaccine boosting. That previous study had a baseline average anti-TT titer of 0.3 IU/mL and then found the mean increase in AB titer after a booster rose to 1.9 IU/mL [13]. Thus, our average baseline TT AB titer of 3.0 IU/mL was 50% greater than their average boosted TT AB titer, which rose 1.6 IU/mL. For anti-drug vaccines baseline pre-vaccination AB titers against methamphetamine, opioids, cocaine, or nicotine would seem unlikely, but is possible, as shown in our previous clinical work with a cocaine vaccine and in other independent studies. We found that before immunization 16% of cocaine-dependent subjects had anti-cocaine IgM levels above 11 µg/mL, which was about one quarter of our target IgG level of 42 ug/mL [15]. All nine of 55 vaccinated subjects with these IgM AB, then had significantly reduced peak IgG anti-cocaine responses after even three booster vaccinations. Thus, in spite of no previous exposure to these anti-drug vaccines, we may have a significant proportion of chronic drug users whom we could predict as poor candidates for anti-drug vaccines, if they have relatively high baseline levels of AB including IgM against the abused drug being targeted by the vaccine.

The ten-fold greater baseline titers of anti-TT and anti-DPT in our study compared to the literature on boosters combined with a strong correlation of the baseline AB titers with the rise in AB after boosting, led us to conduct stratified analyses for TT and DPT AB titers. With this stratification the lower baseline AB group again showed a significant difference in the increase of AB titers between Ent and placebo for DPT and for TT. Significantly more Ent than non-Ent participants (82% vs. 33%) also showed two-fold increases in anti-TT AB titers for this lower baseline TT AB group. The participants with high (above the mean) baseline AB titers showed no difference between Ent and non-Ent, which probably reflected a ceiling effect on how much boosting can be expected to increase these AB titers among individuals who already appear to have adequate AB protection against TT and DPT.

The Ent dose of 1 ug was chosen based on the rodent study showing that the anti-methamphetamine hapten AB peak titer was no higher at 10 ug than at 1 ug Ent, but the anti-TT AB titer was ten-fold larger. This Ent dosage is quite low compared to the doses given safely to humans in previous studies of both healthy subjects and cancer patients, and re-titration of the optimal dose in humans will be part of future studies using the actual anti-drug vaccines rather than just the carrier proteins alone. For example, a larger Ent dose in the current study would probably have substantially enhanced the difference between the Ent and non-Ent groups in anti-TT and anti-DPT titers. However, our goal is ultimately to maximize the AB response against our abused drug hapten and minimize AB against either the carrier protein or the Ent itself. Larger Ent doses could substantially increase AB directed against the Ent, which would be undesirable, particularly since anti-drug vaccination schedules require multiple initial boosters with Ent as well as later boosters beyond 3 to 6 months to extend therapeutic AB titers against the abused drugs. Each of these boosters using Ent could further divert AB production from anti-drug to anti-Ent AB. In this study, we found induction of anti-Ent AB of 3-fold over the baseline levels of anti-Ent AB, with only two of the 25 subjects (8%) having anti-Ent AB levels above 3500 Units, which CBLI considers the threshold for any potential diversion of anti-drug AB production to anti-Ent AB production (personal communication). The baseline anti-Ent AB titers were quite low and probably due to cross-reactivity with AB against flagellin from natural infections earlier in these subjects’ lives. Thus, 8% of potential subjects might make anti-Ent AB, which might neutralize Ent’s efficacy as an adjuvant with subsequent boosters, an undesirable outcome for our planned clinical use of multiple booster vaccinations with an anti-drug vaccine. The effect of anti-Ent AB on anti-drug AB production depends on many factors that would require potentially multiple boosters with Ent to become clinically significant, but the 8% rate was acceptable for further development of Ent as an adjuvant.

The safety data were supportive of clinical use for Ent as a vaccine adjuvant with no serious AEs, and the minor laboratory AEs at six weeks were more frequent in the non-Ent than the Ent group. The possibility of neutralizing anti-Ent AB will need to be examined in future booster studies using multiple vaccinations with Ent itself, but from this study, the increase of 1.6-fold in AB against Ent does not appear to be a serious problem. Furthermore, any baseline titers against flagellin do not appear to have neutralizing cross-reactivity to Ent.

We collected our exploratory genetics data premised on the heritability of antibody responses to tetanus and diphtheria vaccines, which have been estimated at 44% (95% CI: 16–70) and 49% (95% CI: 17–77), respectively, with Toll-like receptor activation contributing an important genetic component of this heritability [16,17]. The data supported our first hypothesis of impairment for eliciting AB titers after vaccination with bacterial proteins in participants with this *TLR5* functional polymorphism compared to “normal” antibody responses to Td vaccine. The “impaired” AG/GG participants had 4-fold greater anti-DPT AB increases with than without Ent, which supported our second hypothesis. Possible explanations for our findings need to consider that the *TLR5* polymorphism *rs2072493* (N592S) is an A to G transversion that codes for the substitution of an asparagine for serine in the C-terminal region of the TLR5 extracellular domain. The variant may affect receptor binding to protein ligands, because of its location in the ligand binding portion of this transmembrane receptor. Although the polymorphism does not affect TLR5 structure, it increases the development of infections by flagellate bacteria, and it may reduce vaccine antibody response to bacterial proteins like DPT more broadly than simply to flagellin [18,19,20,21]. We do not know the mechanism for the benefit from additional stimulation of TLR5 from the Entolimod. This over-stimulation of TLR5 appears to help these individuals match the antibody response that Td vaccine alone could elicit among individuals without this TLR5 receptor “defect. Other types of receptors have shown such receptor compensation by increased numbers of receptors related to genetic polymorphisms that modify the coupling of the receptor to its intracellular effector system or more commonly during chronic pharmacological blockade of a receptor [22,23,24,25].

## 5. Conclusions

In summary, the relatively small number of participants and the high baseline titers of anti-TT and anti-DPT AB limited our ability for showing the efficacy of Ent as an adjuvant that increases AB production. Nevertheless, this study showed a significant enhancement of boosted AB production against DPT and TT and after adjusting for the high baseline titers we enhanced the statistical significance of the increase with Ent compared with the non-Ent group. High anti-Ent AB production is a potential challenge for an estimated 10% of subjects getting this adjuvant repeatedly, and unfortunately, baseline anti-Ent titers were not predictive. Medical safety was excellent, and the pharmacogenetic interaction with the TLR5 functional polymorphism further supported the clinical utility of this adjuvant even for individuals with this relatively common defect in TRL5 functioning.

## Figures and Tables

**Figure 1 vaccines-10-01592-f001:**
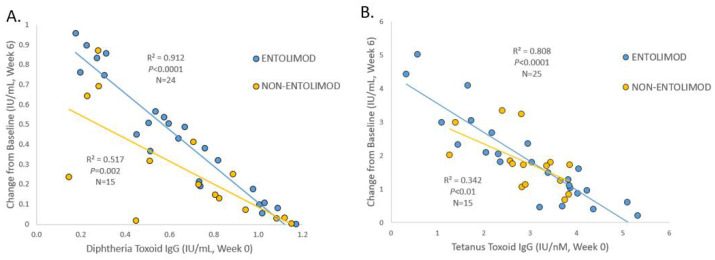
(**A**): Correlation of change in anti-diphtheria antibodies with baseline titers at week 0 for Entolimod and non-Entolimod treatment groups. (**B**): Correlation of change in anti-tetanus antibodies with baseline titers at week 0 for Entolimod and non-Entolimod treatment groups.

## Data Availability

Data may be obtained from the first author TRK upon request from an established investigator with a documented expertise in this area of research without identifiers, if the request is for genetic data on the participants.

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
