# Peer review of "A Clinical Trial of Entolimod a TLR-5 Adjuvant for Vaccines Using Diphtheria or Tetanus as Carrier Proteins"

_vaccines, 2022, doi:10.3390/vaccines10101592_

Round 1

Reviewer 1 Report

Review of vaccines-1898227

Summary. The work focuses on development of vaccines for drugs of abuse by incorporating a new adjuvant (Entolimod, TLR5 agonist). To provide clinical proof of principle for the Entolimod, the authors tested this new adjuvant in a clinical trial combining Entolimod with commercially available vaccines for TT and Diphtheria. The data suggest that Entolimod is a safe and viable adjuvant for further use.

Comment #1. The Introduction is missing an introductory statement that explains the audience what are anti-drug vaccines, and how do they work.

Comment #2. LN42. A statement with references should illustrate which adjuvants have been tested so far with addiction vaccines. Also, a study from the Janda group showed that flagellin, a TLR5 agonist, works with vaccines for drugs of abuse, which supports use of Entolimod.

Comment #3. What is a “normal human”? Healthy human? Please redefine.

Comment #4. LN65: define Td vaccine? Is this Boostrix or a commercial prep of TT/alum?

Comment #5. Can Figure 1 and 2 panels A and B (merge Fig1 A+B, and Fig2 A+B) be merged so that the readers can compare directly the slope of both linear correlations to show the effect of adding Entolimod to TT? The X and Y axes are the same, so it should be pretty straightforward to do this.

Comment #6. LN230, the statement “For anti-drug vaccines this high baseline AB titer is 231

quite unlikely because the participants are highly unlikely to have previous exposure to 232

the methamphetamine, opioid, cocaine, nicotine, or other abused drug haptens in these 233

anti-drug vaccines” Do you expect that people have a baseline drug-specific IgG or IgM antibody or B cell repertoire given the exposure to these drugs. Any clinical or pre-clinical studies to support this statement? If people have higher pre-vaccination frequency or levels of anti-drug antibodies or B cells, would that be detrimental to vaccine responses?

Comment #7. Around LN262 Statements such as “10% of potential subjects might make anti-Ent AB, which is undesirable for anti-hapten AB production” should be elaborate because the authors seemed to assume that Ent may have different effects on TT vs Hapten-TT, on TT- specific vs hapten-specific antibodies and B cells, and that anti-Ent antibodies may affect TT-specific responses but not hapten-specific responses. Also, Ent may enhance TT-specific responses against unconjugated TT in a manner different than Ent effect on TT-specific responses against hapten-TT, which may mask TT epitopes or sites for interactions with T cells.

Comment #8. It would be helpful to explain that hapten-carrier needs activation of both B cells and CD4 T cells to make antibodies. The carrier alone can elicit both B cells and T cells. 

Reviewer 2 Report

This article by Kosten et al. describes a clinical trial using entolimod (TLR5 agonist) as potential adjuvant for anti-drug vaccine using diphtheria and tetanus as carriers. Nevertheless, they do not used entolimod in an antidrug vaccine, rather authors used entolimod in combination with the carrier proteins followed by measurements of antibody titers against each carrier protein and the adjuvant itself. Authors found increase antibody titers against diphtheria and tetanus toxoids following vaccination with entolimod, suggesting its potential use as adjuvant. 

Specific comments:

Since the study does not actually evaluate the use of entolimod with anti-drug vaccine, I would suggest to change the title of the article to one that better reflects the findings.

Using 1ug of adjuvant as it was used in mouse studies seems arbitrary as dosage between mouse and human can be completely different. Therefore, retitration in human hosts might be required to find optimal dosage. Authors should comment on this. 

What about B cells and memory B cell development? would these antibodies last for long enough to confer protection? Authors should comment on this. 

In terms of the role of the TLR5 SNP in the response to entolimod, what is the functional impact of this SNP in TLR5 expression, ligand recognition, signalling and cellular response? How the authors explain the discrepancies regarding antibody response?

Authors refers to some compensatory mechanism by other receptors as the explanation for the increase in antibody titers in TLR5 SNP individuals, what other receptors are they referring to? authors should specify.

Authors should comment more on the cross-reactivity against entolimod and flageling?
